evolution/molecular biology/microbiology

histones, evolution, algae

**Author for correspondence:**
Tobias Warnecke
e-mail: tobias.warnecke@lms.mrc.ac.uk

# Slaying the last unicorn: discovery of histones in the microalga *Nanochlorum eucaryotum*

Valerie W. C. Soo[1,2] and Tobias Warnecke[1,2]

[1]Medical Research Council London Institute of Medical Sciences, London, UK
[2]Institute of Clinical Sciences, Faculty of Medicine, Imperial College London, London, UK

VWCS, 0000-0003-0214-5070; TW, 0000-0002-4936-5428

Histones are the principal constituents of eukaryotic chromatin. The four core histones (H2A, H2B, H3 and H4) are conserved across sequenced eukaryotic genomes and therefore thought to be universal to eukaryotes. In the early 1980s, however, a series of biochemical investigations failed to find evidence for histones or nucleosomal structures in the microscopic green alga *Nanochlorum eucaryotum*. If true, derived histone loss in this lineage would constitute an exceptional case that might help us further understand the principles governing eukaryotic gene regulation. To substantiate these earlier reports of histone loss in *N. eucaryotum*, we sequenced, assembled and quantified its transcriptome. Following a systematic search for histone-fold domains in the assembled transcriptome, we detect orthologues to all four core histones. We also find histone mRNAs to be highly expressed, comparable to the situation in other eukaryotes. Finally, we obtain characteristic protection patterns when *N. eucaryotum* chromatin is subjected to micrococcal nuclease digestion, indicating widespread formation of nucleosomal complexes *in vivo*. We conclude that previous reports of missing histones in *N. eucaryotum* were mistaken. By all indications, *Nanochlorum eucaryotum* has histone-based chromatin characteristic of most eukaryotes.

## 1. Introduction

Nucleosomes are the fundamental repeat units of eukaryotic chromatin; four core histones (H3, H4, H2A and H2B) assembled into octameric complexes that wrap approximately 150 bp of DNA. All four core histones were present in the last common ancestor of eukaryotes [1,2], have since been retained along distant eukaryotic lineages, and are considered part of the universal, indispensable molecular toolkit of eukaryotes.

This was not always the case. In the pre-genomic era, histones were at one point thought to be absent, for example, from the chromatin of several fungi including *Neurospora crassa* [3,4], *Allomyces arbusculus* [5] and *Phycomyces blakesleeanus* [4]. The chromatin of dinoflagellates was also considered free of histones [6].

Early claims of missing histones in fungi did not stand up to closer scrutiny: improved biochemical protocols soon found histones in high abundance [7,8]. The case of dinoflagellates proved more complicated. Histones do indeed seem to have lost their role as the principal packaging agent of dinoflagellate DNA. Other small basic proteins organize their permanently condensed chromatin [9–11]. Sequencing of multiple dinoflagellate genomes, however, revealed full complements of core histone genes [12], which appear to be lowly expressed and whose functional roles remain unclear [9].

We are aware of one other putative case of histone loss in eukaryotes. In a series of papers in the early 1980s, histones and nucleosomal structures were reported absent from the microscopic green alga *Nanochlorum eucaryotum* [13,14]. This claim has gone unchallenged since and been reiterated decades later [15,16] as a potential example of deviant eukaryotic chromatin architecture.

*Nanochlorum eucaryotum* belongs to the Trebouxiophyceae [17–20], which also comprise more intensively studied species of the genus *Chlorella*. Isolated from a seawater aquarium that contained material originally sampled off Rovinj, Croatia, *N. eucaryotum* was described as minute in size (approx. 1–2 µm in diameter), with a typical 1 : 1 : 1 relationship between nucleus, chloroplast and mitochondrion [13]. Its chromatin appeared decondensed throughout a closed mitosis. And, despite intensive efforts to detect histones and nucleosomal structures by biochemical means, electron microscopy and comparative hybridization, neither histones nor the classic beads-on-a-string arrangement of nucleosomes along DNA were observed [13,14].

Here, we sequence the transcriptome of *N. eucaryotum* to show that histones are present, highly expressed, and closely related to histone proteins in other members of the Trebouxiophyceae. We also find evidence to support nucleosome formation, as micrococcal nuclease (MNase) digestion reveals characteristic nucleosomal ladder patterns. Based on this evidence we conclude that *N. eucaryotum* is not a unicorn.

# 2. Results and discussion

To establish whether histones are present in *N. eucaryotum* we obtained strain SAG 55.87 from the Culture Collection of Algae at the University of Göttingen (Sammlung von Algenkulturen der Universität Göttingen, SAG). This strain is identical to strain UTEX 2502 in the Culture Collection of Algae at the University of Texas at Austin [20], even though the associated species designations are not. Early comparative analyses, both phylogenetic and morphological, triggered a veritable bonanza of valid and invalid taxonomic revisions, as *Nanochlorum eucaryotum* became variously known as *Nannochloris eucaryotum* [21], the name still associated with UTEX 2502, and *Pseudochloris wilhelmii*, the name currently associated with SAG 55.87 [20]. Below, to avoid confusion, we will use the term *Nanochlorum eucaryotum* (SAG 55.87) to refer to our results. SAG 55.87 is the strain on which the pertinent observations about histones were made and which was deposited in SAG by the original authors (C. Wilhelm 2020, personal communication).

## 2.1. Detection of histone transcripts in Nanochlorum eucaryotum (SAG 55.87)

At the outset of this study, no genome assembly or functional genomic data were available for SAG 55.87. To search for evidence of histones and assess their likely role in the formation of *N. eucaryotum* (SAG 55.87) chromatin, we therefore extracted RNA from a 31-day-old *N. eucaryotum* (SAG 55.87) culture and assembled the algal transcriptome de novo (see Methods).

We then screened 6-frame translations of all assembled transcripts for histone domains using hidden Markov models from Pfam (see Methods). This search revealed histone domains in several predicted transcripts. Following manual curation, for example to only retain the longest unique polypeptides (see Methods), we used BLAST (blastp) to identify, in an unbiased manner, proteins with the highest similarity to the candidate histones in the non-redundant NCBI database. The top hits for several *N. eucaryotum* (SAG 55.87) candidate histones were annotated core histones from previously sequenced members of the Trebouxiophyceae (including *Auxenochlorella protothecoides*, *Micratinium conductrix* and *Chlorella variabilis*, electronic supplementary material, table S1). We also classified candidate histones using the Histone Variants Database (HistoneDB 2.0) and found that the strongest similarities were to canonical H3, canonical H4, canonical H2B and H2A.Z (electronic supplementary material, table S1).

Using the same domain scanning approach, we also found putative orthologues for tubulin (Pfam domain PF00091.25), another protein initially believed to be absent because *N. eucaryotum* (SAG 55.87) appeared insensitive to mitotic spindle inhibitors [14].

To place candidate histone hits in phylogenetic context, we added putative *N. eucaryotum* (SAG 55.87) histones to a large prior alignment of eukaryotic and archaeal histones [22] and subsequently built a phylogenetic tree (see Methods) from *N. eucaryotum* (SAG 55.87) candidate histones and the histones of *C. variabilis* NC64, for which a high-quality genome assembly is available. We find candidate *N. eucaryotum* (SAG 55.87) histones that have high similarity to and cluster with *C. variabilis* H2A, H2B, H3 and H4 (figure 1; electronic supplementary material, figure S1). In some instances, we can assign candidate *N. eucaryotum* (SAG 55.87) histones to specific variants, notably H2A.Z and H3.1 (electronic supplementary material, figure S1). However, we note that our analysis—which does not involve genome assembly—was not geared towards comprehensive identification of histone variants in the *N. eucaryotum* (SAG 55.87) genome. As a consequence, we cannot state with confidence whether particular variants (e.g. H2A.X) are absent from the genome or simply missing from our transcriptome assembly. The affiliation between *C. variabilis* H4 and its likely *N. eucaryotum* (SAG 55.87) orthologues is less tight than for the other core histones and the putative *N. eucaryotum* (SAG 55.87) orthologue appears unusually divergent. However, it is still more similar to canonical H4 than to other histone proteins (figure 1; electronic supplementary material, figure S1 and table S1).

We conclude that the *N. eucaryotum* (SAG 55.87) genome encodes and expresses a full complement of core histone proteins.

## 2.2. Histone transcripts in *Nanochlorum eucaryotum* (SAG 55.87) are abundant

Although we detect all four core histones in the *N. eucaryotum* (SAG 55.87) transcriptome, it is conceivable that their abundance is low, precluding formation of nucleosomes genome-wide, in a manner reminiscent of dinoflagellates [9]. This would be consistent with earlier failed attempts to detect histones by biochemical means [13,14].

To assess the abundance of histones, and therefore their likely capacity to act as global organizers of *N. eucaryotum* (SAG 55.87) chromatin, we quantified the relative abundance of reconstructed transcripts, including those that contain histone domains (see Methods). We then considered two metrics: the overall histone expression level (by summing abundance over all relevant transcripts) and the individual histone transcript with the highest expression level. On both counts, *N. eucaryotum* (SAG 55.87) histones must be considered highly expressed when compared to non-histone transcripts (figure 2). To compare relative histone abundance in *N. eucaryotum* (SAG 55.87) with other species, we de novo-assembled transcriptomes of five other eukaryotes, including three algae, from publicly available RNA-Seq data (see Methods). To ensure comparability, raw reads were processed, assembled and quantified following the same pipeline used for *N. eucaryotum* (SAG 55.87). As expected, histones are highly expressed in all eukaryotes tested. More importantly, there is no indication (figure 2) that histones are relatively less abundant in *N. eucaryotum* (SAG 55.87).

We conclude that histone genes are likely to be present in sufficient abundance in *N. eucaryotum* (SAG 55.87) to mediate typical eukaryotic chromatin organization.

## 2.3. Evidence for widespread nucleosome formation in *Nanochlorum eucaryotum* (SAG 55.87)

Even though histones are present and highly expressed it is still conceivable that they fail to form nucleosomal complexes. Nucleosomes in other eukaryotes provide protection from MNase digestion, which produces characteristic ladder patterns when the resulting DNA fragments are separated on a gel, with individual rungs of the ladder corresponding to fragments protected by one or consecutively more nucleosomes.

To establish whether nucleosomal complexes form in *N. eucaryotum* (SAG 55.87), we therefore carried out a series of MNase digestion experiments. We find that MNase digestion of *N. eucaryotum* (SAG 55.87) chromatin produces a ladder pattern indistinguishable from patterns observed in other eukaryotes (figure 3). We conclude that widespread nucleosome formation occurs in *N. eucaryotum* (SAG 55.87).

In summary, we detect highly abundant mRNAs corresponding to the four core histones in the transcriptome of *N. eucaryotum* (SAG 55.87), which are similar in sequence to the histones of closely related green algae. In all aspects we have investigated, including the formation of nucleosomes inferred from MNase digestion, we find no indication that *N. eucaryotum* (SAG 55.87) is unusual when compared with other eukaryotes. While eukaryotes without histones might yet exist, they remain to be found.

R. Soc. Open Sci. **8**: 202023

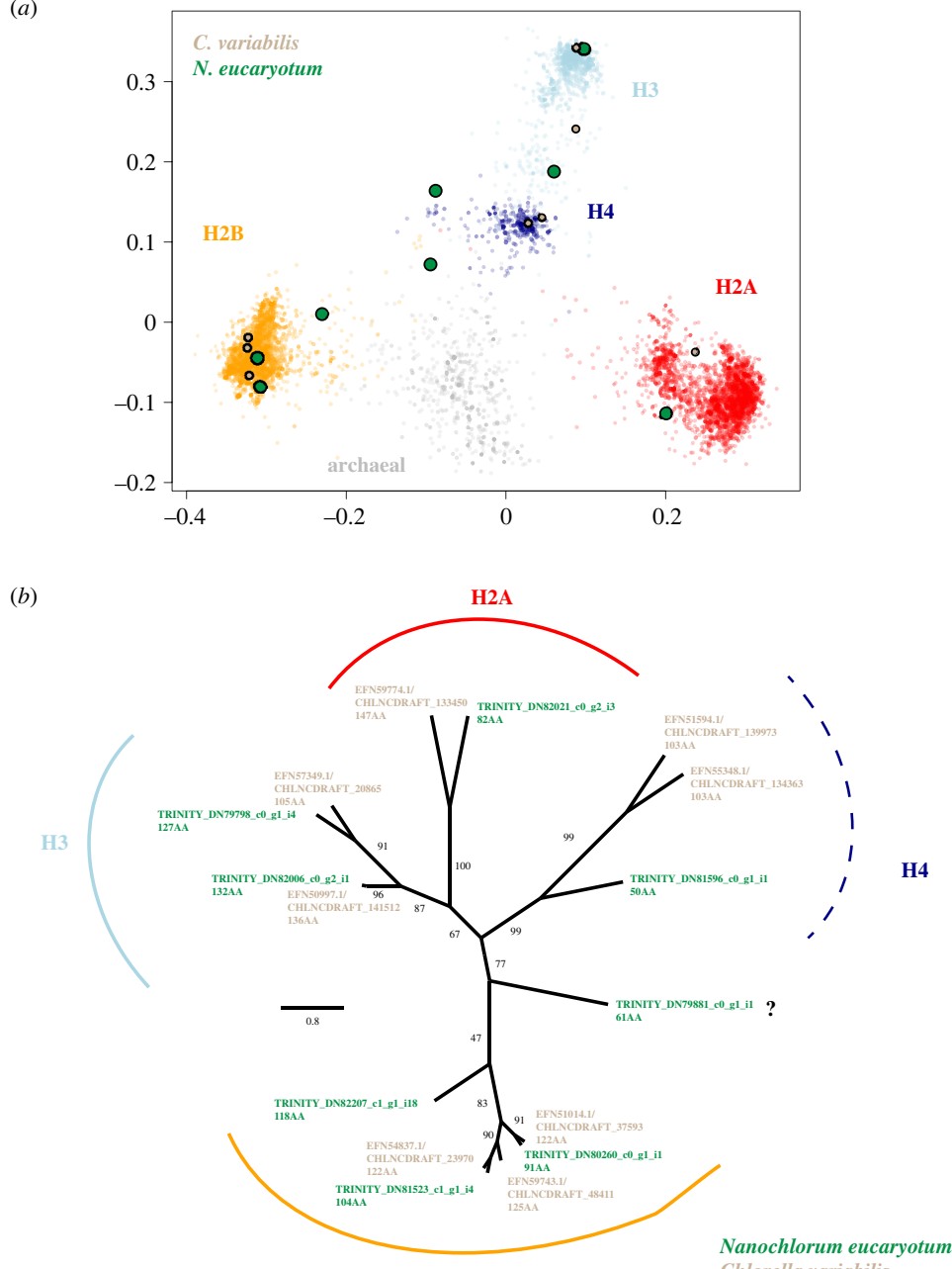

**Figure 1.** *Nanochlorum eucaryotum* (SAG 55.87) candidate histones in phylogenetic context. (*a*) Similarity of eukaryotic and archaeal histones based on classic multidimensional scaling, which renders distance relationships (derived from the alignment) in two dimensions. Histones annotated as H2A, H2B, H3, H4 or of archaeal origin are represented by distinct colours and fall into reasonably distinct clusters. Histones from *C. variabilis* and histone candidates from *N. eucaryotum* (SAG 55.87) associate with each of the eukaryotic clusters, suggesting that orthologues for all four core histones are present in these species. The alignment underlying this analysis is provided in Stevens *et al.* [22]. (*b*) Maximum-likelihood phylogenetic tree of *C. variabilis* and *N. eucaryotum* (SAG 55.87) histones. The length of the relevant polypeptides is provided in amino acids (AA) for each sequence. *Nanochlorum eucaryotum* (SAG 55.87) candidate histones are represented by IDs from the *de novo* transcript assembly (see electronic supplementary material, table S1). A single reconstructed peptide whose affiliation to either H4 or H2B is uncertain is marked with a question mark.

# 3. Methods

## 3.1. Algal strain and growth conditions

*Pseudochloris wilhelmii/Nanochlorum eucaryotum* strain 55.87 was obtained from the Culture Collection of Algae at the University of Göttingen (Sammlung von Algenkulturen der Universität Göttingen, SAG)

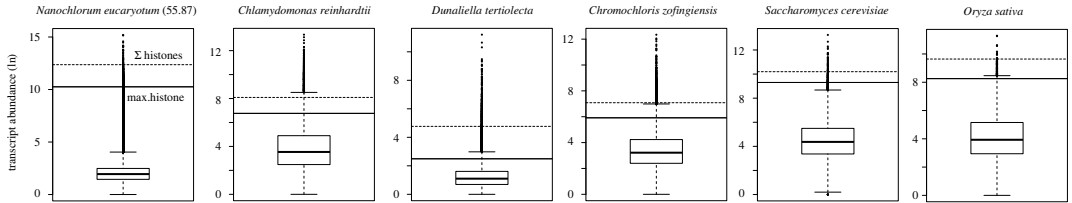

**Figure 2.** Abundance of histone transcripts compared with the remainder of de novo-assembled transcripts in *N. eucaryotum* (SAG 55.87) and five other eukaryotes, including three algae (*C. reinhardtii*, *D. tertiolecta* and *C. zofingiensis*). Abundance of the most highly expressed histone transcript (solid lines) and the summed abundance of all histone transcripts (dotted lines) is indicated for each species. See Methods for details on the underlying RNA-Seq data, assembly and quantification process.

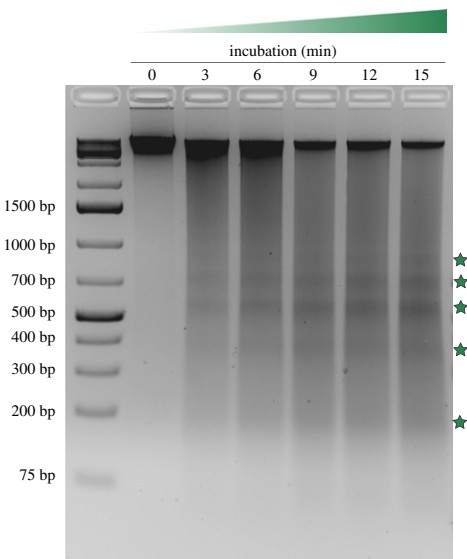

**Figure 3.** MNase digestion of *N. eucaryotum* (SAG 55.87) chromatin. Discrete DNA fragments (green stars) are indicative of nucleosomal protection from MNase digestion. Each reaction shown in this agarose gel was carried out with 90 units of MNase and approximately $10^7$ algal cells.

and grown in liquid culture at room temperature near a window that receives 8–12 h of indirect sunlight each day. Algal growth was monitored microscopically using direct cell counts derived from a haemocytometer. As recommended by SAG, SAG 55.87 was grown in Brackish Water Medium (200 mg l$^{-1}$ KNO$_3$, 20 mg l$^{-1}$ K$_2$HPO$_4$, 20 mg l$^{-1}$ MgSO$_4$, 4 mg l$^{-1}$ EDTA, 3.5 mg l$^{-1}$ FeSO$_4$, 0.05 mg l$^{-1}$ H$_3$BO$_3$, 0.01 mg l$^{-1}$ MnSO$_4$, 0.005 mg l$^{-1}$ ZnSO$_4$, 0.005 mg l$^{-1}$ Co(NO$_3$)$_2$, 0.005 mg l$^{-1}$ Na$_2$MoO$_4$, 2.5 × 10$^{-5}$ mg l$^{-1}$ CuSO$_4$, 0.005 mg l$^{-1}$ vitamin B$_{12}$, 45.5% (v/v) 0.2 µm-filtered natural seawater (SeaWater NSW) and 3% (v/v) autoclaved soil extract (Canna Terra Professional Soil Mix; part no. 02–075-050)). All cultures were sealed with parafilm to prevent moisture loss during incubation.

## 3.2. Total RNA extraction

Total RNA was isolated from a 31-day-old *N. eucaryotum* (SAG 55.87) culture consisting of approximately $1.9 \times 10^7$ cells in total. To minimize the risk of bacterial contamination, 0.2 µm-filtered lysozyme solution was added into the culture to a final concentration of 0.5 mg ml$^{-1}$ 2 days prior to cell harvest. After harvesting the culture by centrifugation (13 000 × *g*, fixed-angle rotor, 5 min, 4°C), the resulting cell pellet was resuspended in 1 ml of RNA Protection Buffer (1 × NEB DNA/RNA Protection Reagent, 1% (w/v) polyvinylpyrrolidone-40, 0.3% (v/v) beta-mercaptoethanol), and was split into two 0.5 ml aliquots. An equal volume (0.5 ml) of autoclaved acid-washed glass beads (425–600 µm diameter) was added into each cell suspension aliquot, and the mixture was mechanically lysed in a Qiagen TissueLyser II at 20 Hz for five cycles of 1 min on/1 min off. Next, cellular debris was pelleted (13 000 × *g*, 2 min, 4°C), and the supernatant was transferred into a new microcentrifuge tube containing an equal volume of NEB RNA Lysis Buffer. Following a thorough mixing, the mixture was applied to a NEB gDNA Removal Column to remove genomic DNA. Total RNA was subsequently column-purified as described in the NEB

Monarch Total RNA Miniprep Kit, and eluted using 60 µl nuclease-free water. Two technical replicates were carried out for RNA extractions. These were multiplexed for sequencing and reads pooled prior to processing and assembly (see below).

## 3.3. RNA sequencing

RiboMinus Plant Kit for RNA-Seq (Invitrogen) was used to deplete ribosomal RNAs (rRNAs) from the two RNA samples. The rRNA-depleted RNA samples were then used for library preparation (RNA fragmentation, cDNA synthesis, adaptor ligation and indexing) following the protocol of the NEBNext Ultra II Directional RNA Library Prep Kit for Illumina. The final libraries were subjected to 100 bp paired-end sequencing on an Illumina Hiseq2500 v. 4 sequencer. Library quality and yield were assessed on a Bioanalyser 2100 and Qubit fluorometer, respectively. RNA sequencing data have been deposited in the NCBI Sequence Read Archive with accession PRJNA670301 (SRR12853754 for technical replicate 1, SRR12853753 for technical replicate 2).

## 3.4. Genomic DNA extraction

As part of this project, we also extracted and sequenced genomic DNA from *N. eucaryotum* (SAG 55.87). We do not analyse or discuss DNA sequencing results above but document below how these data were derived so that they can be re-used by others in the future.

Genomic DNA extraction was carried out as described by [23] with minor modifications. A lysozyme-treated *N. eucaryotum* culture (corresponding to approx. $7.4 \times 10^7$ cells) was harvested by centrifugation ($2500 \times g$, swing-bucket rotor, 20 min, 4°C). The cell pellet was then washed thrice with TE buffer (10 mM Tris–HCl pH 7.5, 0.1 mM EDTA) to remove as much growth medium as possible. Next, the cell pellet was resuspended in 1.5 ml Extraction Buffer (2% (v/v) Triton X-100, 1% (v/v) SDS, 100 mM NaCl, 10 mM Tris–HCl pH 8.0, 1 mM EDTA) and split into two 0.75 ml aliquots. After combining each aliquot with an equal volume of autoclaved acid-washed glass beads (425–600 µm diameter), the cell mixture was mechanically lysed in a Qiagen TissueLyser II as described above (Total RNA extraction). After pooling the lysate (approx. 1.5 ml), proteinase K was added to the lysate to a final concentration of 0.15 mg ml$^{-1}$, and the mixture was incubated at 55°C for 1 h to allow digestion of cellular proteins. Further added to the mixture were 0.75 ml of 5 M NaCl and 0.4 ml of pre-warmed CTAB buffer (1% CTAB, 1.4 M NaCl, 100 mM Tris–HCl pH 8.0, 10 mM EDTA). After a 10 min incubation at 65°C, nucleic acids were extracted using phenol : chloroform : isoamyl alcohol (25 : 24 : 1) three times, and were precipitated with 0.1 volume of 3M sodium acetate (pH 5.5) and 2 volumes of 100% ice-cold ethanol. Following 1 h of precipitation at −20°C, nucleic acids were centrifuged at $21\,000 \times g$ for 30 min at 4°C. The resulting pellet was air-dried, solubilized in 500 µl TE buffer containing 0.12 mg ml$^{-1}$ RNase A, and incubated at 37°C for 2.5 h to allow digestion of cellular RNA. Total genomic DNA was re-extracted and re-precipitated as before. The final DNA pellet was air-dried, and then re-solubilized in 100 µl of 10 mM Tris–HCl (pH 8.0).

## 3.5. Genome sequencing

Algal genomic DNA was sequenced on two platforms: Oxford Nanopore and Illumina Hiseq. For the former platform, 500 ng of genomic DNA was used to construct the sequencing library according to the protocol of the Ligation Sequencing Kit (Oxford Nanopore Technologies). The sequencing library was then sequenced on an Oxford Nanopore Optimised Flongle Flow Cell for a period of 24 h using the MINKNOW software application. For the Illumina platform, 200 ng of the genomic DNA (from the same batch) was used to prepare the sequencing library (400–500 bp) following the recommended protocol of the NEBNext Ultra II FS DNA Library Prep Kit for Illumina. The final library was subjected to 100 bp paired-end sequencing on an Illumina Hiseq2500 v. 4 sequencer. Genomic DNA and sequencing libraries were quantified using Qubit fluorometric assays while DNA quality was assessed on a Bioanalyser 2100. Sequencing data for genomic DNA have been deposited in the NCBI Sequence Read Archive under accession PRJNA670458.

## 3.6. *In situ* MNase digestion of algal chromatin

*Nanochlorum eucaryotum* (SAG 55.87) chromatin was digested with MNase as described by Potdar *et al.* [24]. Briefly, up to $9 \times 10^7$ *N. eucaryotum* (SAG 55.87) cells were harvested by centrifugation ($2500 \times g$,

swing-bucket rotor, 20 min, 4°C), washed once with TMC buffer (50 mM Tris–HCl pH 7.5, 5 mM $MgCl_2$, 5 mM $CaCl_2$) and then resuspended in 1 ml of TMC buffer. Next, the cell suspension was divided into 0.2 ml aliquots, each of which was added into a 2 ml microcentrifuge tube containing 300 mg sterile acid-washed glass beads (425–600 µm diameter). After adding an appropriate amount of MNase (ThermoFisher Scientific; approx. 90 units per reaction), the mixture was immediately vortexed for 1 min at the highest setting and was incubated at 37°C for at least 3 min. To halt enzymatic digestion, 20 µl of 10 × STOP buffer (200 mM EDTA and 5% (v/v) SDS) was added into each mixture. The cell lysate was recovered and transferred into a new microcentrifuge tube. To recover the nucleic acid fraction, an equal volume of 2 × CTAB buffer (200 mM Tris–HCl pH 8, 40 mM EDTA, 2.8 M NaCl, 4% CTAB) was added into the cell lysate and the mixture was allowed to incubate at 65°C for 1 h. Nucleic acid extraction was subsequently carried out using phenol : chloroform : isoamyl alcohol (25 : 24 : 1) as described above (Genomic DNA extraction). The resulting nucleic acid fraction was treated with 200 mg RNase A at 37°C for 2 h, and the algal DNA was finally cleaned up by re-extraction using phenol : chloroform : isoamyl alcohol (25 : 24 : 1). To view the extent of MNase digestion of the algal chromatin, the extracted DNA was electrophoresed on 3% agarose gels at 80 V for 1 h.

### 3.7. *De novo* transcriptome assembly

Reads from the two technical replicates (see above) were combined and trimmed using TrimGalore 0.6.5 (-q 30 –illumina –paired -trim1 –gzip –length 80) with cutadapt v. 2.8 to remove adapters and low-quality bases.

*De novo* transcriptomes were assembled using Trinity (v. 2.4.0) and quantified using kallisto (v. 0.46.1; parameters -b 100 for paired-end data and -b 100 -l 180 -s 20 for single-end data).

Comparative transcriptomic datasets were identified by searching NCBI GEO for Illumina (Tru-Seq)-based rRNA-depleted RNA-seq data. Reads were trimmed and transcriptomes de novo assembled and quantified as above and transcripts containing histone folds identified as detailed below. Accession numbers for these datasets, represented in figure 2, are as follows: *Saccharomyces cerevisiae* (GEO accession: GSM1892898; SRA accession: SRR2517449); *Oryza sativa Japonica* (GEO accession: GSM1585988; SRA accession: SRR1761780); *Chlamydomonas reinhardtii* (GEO accession: GSM3069474; SRA accession: SRR6904722); *Dunaliella tertiolecta* (GEO accession: GSM1821015; SRA accession: SRR2099913); *Chromochloris zofingiensis* (GEO accession: GSM2431131; SRA accession: SRR5117383).

### 3.8. Detection of histone mRNAs

Predicted transcripts were 6-frame translated into a polypeptide library. The standard genetic code was used for translation as we did not expect histones to be encoded by either the chloroplast or mitochondrion. Whenever a stop codon occurred, the upstream and downstream peptides were considered separate library entries. Peptides shorter than 30 amino acids were discarded. This predicted polypeptide library was then searched for hits against three Pfam domain models: PF00125 (comprising all four eukaryotic core histones), PF00538 (H1 linker histone) and PF00808 (archaeal histones/nuclear factor Y) using hmmsearch from the HMMER package (v. 3.3). Hits with an E-value <0.001 were considered further and filtered for the longest unique polypeptides.

### 3.9. Phylogenetic analysis

The remaining set of candidate histone-fold proteins were added to a pre-existing alignment of eukaryotic and archaeal histone-fold proteins [22] using Mafft v. 7.310 (-add -reorder) along with annotated histones from *C. variabilis* NC64A obtained from Uniprot. A sub-alignment of candidate *N. eucaryotum* (SAG 55.87) and *C. variabilis* NC64A histones was extracted and used to build a maximum-likelihood phylogenetic tree using RAxML (v. 8.1.16 -f a -m PROTCATAUTO -N 100) with 100 rapid bootstraps [25].

Data accessibility. All sequencing data have been deposited in the NCBI Sequence Read Archive under accessions PRJNA670301 and PRJNA670458.

Authors' contributions. V.W.C.S. cultured the algae, carried out RNA/DNA extractions and MNase digestion and contributed to writing the manuscript. T.W. carried out bioinformatic and phylogenetic analyses and contributed to writing the manuscript.

Competing interests. We declare we have no competing interests.

Funding. We received no funding for this study.

Acknowledgements. This work was supported by Medical Research Council core funding to T.W. We thank members of the Molecular Systems group and Christian Wilhelm for discussions, the Sammlung von Algenkulturen der Universität Göttingen (SAG) for providing strain SAG 55.87, the LMS Genomics facility for library preparation and sequencing and John West for advice on algal culture.

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
