## [Peer Review File · Royal Society Open Science]

Review History

RSOS-202023.R0 (Original submission)

Review form: Reviewer 1

Is the manuscript scientifically sound in its present form?

Yes

Are the interpretations and conclusions justified by the results?

Yes

Is the language acceptable?

Yes

Do you have any ethical concerns with this paper?

No

Have you any concerns about statistical analyses in this paper?

No

Recommendation?

Major revision is needed (please make suggestions in comments)

Comments to the Author(s)

The manuscript by Soo & Warnecke. investigates whether histones are present in *Nanochlorum* eucaryotum, an extremely small unicellular green alga that has been reported to lack histones. The authors assemble transcriptomes from *Nanochlorum* and related green algae and show clearly that *Nanochlorum* has at least one histone paralog in each of the major classes of H2A, H2B, H3, and H4, and they are abundantly transcribed, similar to other eukaryotes. They show by micrococcal nuclease assay that the expected ladder sizes attributable to nucleosomes are present, implying that the histones are assembled into nucleosomal chromatin.

This is an interesting addendum to the universality of chromatin composition in eukaryotes that resolves an outstanding apparent exception. My only disappointment is that the authors fail to do even the most rudimentary bioinformatic assessment of the histones they uncover. For example, nearly all eukaryotes have at least two paralogs of H2A, one that is an ortholog of H2A.Z and one or more that is a paralog of H2A/H2A.X. Peculiarly, *Nanochlorum* eucaryotum and its relative *Chlorella variabilis* appear to have only a single H2A. Are these H2As identifiable as H2A.Z-like or H2A.X-like? H2A.X in other organisms is readily identifiable by its C-terminal phosphorylation motif SQE Φ , where Φ is a hydrophobic residue, often F or Y. H2A.Z can be identified not only by conserved residues, especially the extended acidic patch, but even in the divergent H2As of trypanosomes by a one amino acid insertion in Loop1 and a one amino acid deletion in the C-terminal docking domain relative to other H2As. Is the *Nanochlorum* H2A identifiable as either of these conserved paralogs? It would be of interest to know whether it is missing one or both of them.

Similarly, *Nanochlorum* and *Chlorella* have two H3s each. Unicellular eukaryotes often have an H3.3-like paralog, and most have a cenH3 paralog. While functional assays for cenH3 are presumably beyond the scope of the present manuscript, bioinformatic criteria (long tail, longer Loop 1, and only about 50% identity to conventional H3s) can identify putative cenH3s. This is relevant to “slaying unicorns” because Zahn reported no sign of mitosis or meiosis, whereas a putative cenH3 would argue for the likelihood of these processes. As a side note, Zahn also reported a lack of microtubules identifiable by inhibitors, and transcripts for tubulins are likely present in the transcriptomes the authors have assembled.

Having three H2Bs for a single H2A is unusual, and while it may be impossible to assess from bioinformatics what their roles are, It would be interesting to see in an alignment how they differ from each other. Similarly the divergent H4s or H4-like histones are unusual in that H4 is ordinarily one of the most strongly conserved proteins known. Alignments might reveal interesting features, or add to the mystery of their specialization. All four classes of histones typically have conserved sites of modification, especially lysines that can be acetylated, methylated or ubiquitylated. Are these conserved residues present in the various paralogs the authors have discovered? Are there one or more H1 paralogs? These questions can presumably be answered relatively easily with the data in hand, and would make for a more interesting paper. Comparison with *Chlorella* and the other green algal transcriptomes that the authors have assembled might offer some insight into the evolution of *Nanochlorum*'s unique histone complement.

Minor comment:

Figure 1 Legend: It would be helpful if the authors could elaborate a bit on what they mean by “classical multidimensional scaling”. What are the dimensions, or how should I understand this?

Decision letter (RSOS-202023.R0)

Dear Dr Warnecke

On behalf of the Editors, we are pleased to inform you that your Manuscript RSOS-202023 "Slaying the last unicorn - discovery of histones in the microalga *Nanochlorum eucaryotum*" has been accepted for publication in Royal Society Open Science subject to minor revision in accordance with the referees' reports. Please find the referees' comments along with any feedback from the Editors below my signature.

Please submit your revised manuscript and required files (see below) no later than 7 days from today's (ie 11-Jan-2021) date. Note: the ScholarOne system will 'lock' if submission of the revision is attempted 7 or more days after the deadline. If you do not think you will be able to meet this deadline please contact the editorial office immediately.

on behalf of Dr Ed Bolt (Associate Editor) and Kevin Padian (Subject Editor)
openscience@royalsociety.org

Associate Editor Comments to Author (Dr Ed Bolt):

Associate Editor: 1

Comments to the Author:

Thanks for submitting this manuscript. In agreement with the reviewer, I'd be pleased to recommend that the paper is accepted if you could include some amino acid sequence alignment information, and possibly too some simple structural modelling/comparison.

best,
Ed

Reviewer comments to Author:

Reviewer: 1

Comments to the Author(s)

The manuscript by Soo & Warnecke. investigates whether histones are present in *Nanochlorum* eucaryotum, an extremely small unicellular green alga that has been reported to lack histones. The authors assemble transcriptomes from *Nanochlorum* and related green algae and show clearly that *Nanochlorum* has at least one histone paralog in each of the major classes of H2A, H2B, H3, and H4, and they are abundantly transcribed, similar to other eukaryotes. They show by micrococcal nuclease assay that the expected ladder sizes attributable to nucleosomes are present, implying that the histones are assembled into nucleosomal chromatin.

This is an interesting addendum to the universality of chromatin composition in eukaryotes that resolves an outstanding apparent exception. My only disappointment is that the authors fail to do even the most rudimentary bioinformatic assessment of the histones they uncover. For example, nearly all eukaryotes have at least two paralogs of H2A, one that is an ortholog of H2A.Z and one or more that is a paralog of H2A/H2A.X. Peculiarly, *Nanochlorum* eucaryotum and its relative *Chlorella variabilis* appear to have only a single H2A. Are these H2As identifiable as H2A.Z-like or H2A.X-like? H2A.X in other organisms is readily identifiable by its C-terminal phosphorylation motif SQE Φ , where Φ is a hydrophobic residue, often F or Y. H2A.Z can be identified not only by conserved residues, especially the extended acidic patch, but even in the divergent H2As of trypanosomes by a one amino acid insertion in Loop1 and a one amino acid deletion in the C-terminal docking domain relative to other H2As. Is the *Nanochlorum* H2A identifiable as either of these conserved paralogs? It would be of interest to know whether it is missing one or both of them.

Similarly, *Nanochlorum* and *Chlorella* have two H3s each. Unicellular eukaryotes often have an H3.3-like paralog, and most have a cenH3 paralog. While functional assays for cenH3 are presumably beyond the scope of the present manuscript, bioinformatic criteria (long tail, longer Loop 1, and only about 50% identity to conventional H3s) can identify putative cenH3s. This is relevant to “slaying unicorns” because Zahn reported no sign of mitosis or meiosis, whereas a putative cenH3 would argue for the likelihood of these processes. As a side note, Zahn also reported a lack of microtubules identifiable by inhibitors, and transcripts for tubulins are likely present in the transcriptomes the authors have assembled.

Having three H2Bs for a single H2A is unusual, and while it may be impossible to assess from bioinformatics what their roles are, It would be interesting to see in an alignment how they differ from each other. Similarly the divergent H4s or H4-like histones are unusual in that H4 is ordinarily one of the most strongly conserved proteins known. Alignments might reveal interesting features, or add to the mystery of their specialization. All four classes of histones typically have conserved sites of modification, especially lysines that can be acetylated, methylated or ubiquitylated. Are these conserved residues present in the various paralogs the authors have discovered? Are there one or more H1 paralogs? These questions can presumably be answered relatively easily with the data in hand, and would make for a more interesting paper. Comparison with *Chlorella* and the other green algal transcriptomes that the authors have assembled might offer some insight into the evolution of *Nanochlorum*'s unique histone complement.

Minor comment:

Figure 1 Legend: It would be helpful if the authors could elaborate a bit on what they mean by “classical multidimensional scaling”. What are the dimensions, or how should I understand this?

===PREPARING YOUR MANUSCRIPT===

===PREPARING YOUR REVISION IN SCHOLARONE===

- An individual file of each figure (EPS or print-quality PDF preferred [either format should be produced directly from original creation package], or original software format).
 - An editable file of each table (.doc, .docx, .xls, .xlsx, or .csv).
 - An editable file of all figure and table captions.
- Note: you may upload the figure, table, and caption files in a single Zip folder.
- Any electronic supplementary material (ESM).
 - If you are requesting a discretionary waiver for the article processing charge, the waiver form must be included at this step.
 - If you are providing image files for potential cover images, please upload these at this step, and inform the editorial office you have done so. You must hold the copyright to any image provided.
 - A copy of your point-by-point response to referees and Editors. This will expedite the preparation of your proof.

- Ensure that your data access statement meets the requirements at <https://royalsociety.org/journals/authors/author-guidelines/#data>. You should ensure that you cite the dataset in your reference list. If you have deposited data etc in the Dryad repository, please only include the 'For publication' link at this stage. You should remove the 'For review' link.
- If you are requesting an article processing charge waiver, you must select the relevant waiver option (if requesting a discretionary waiver, the form should have been uploaded at Step 3 'File upload' above).
- If you have uploaded ESM files, please ensure you follow the guidance at <https://royalsociety.org/journals/authors/author-guidelines/#supplementary-material> to include a suitable title and informative caption. An example of appropriate titling and captioning may be found at https://figshare.com/articles/Table_S2_from_Is_there_a_trade-off_between_peak_performance_and_performance_breadth_across_temperatures_for_aerobic_scops_in_teleost_fishes_/3843624.

Author's Response to Decision Letter for (RSOS-202023.R0)

See Appendix A.

Decision letter (RSOS-202023.R1)

Dear Dr Warnecke,

It is a pleasure to accept your manuscript entitled "Slaying the last unicorn - discovery of histones in the microalga *Nanochlorum eucaryotum*" in its current form for publication in Royal Society Open Science.

on behalf of Dr Ed Bolt (Associate Editor) and Kevin Padian (Subject Editor)
openscience@royalsociety.org

Associate Editor Comments to Author (Dr Ed Bolt):
Associate Editor
Comments to the Author:
Addresses comments raised by external reviewer - additional material suitable.

Appendix A

Point-by-point response to reviewer comments

Editor

1. *In agreement with the reviewer, I'd be pleased to recommend that the paper is accepted if you could include some amino acid sequence alignment information, and possibly too some simple structural modelling/comparison.*

We now include amino acid alignments generated with MAFFT-linsi for all four histones (H2A, H2B, H3, H4) and highlight affiliation of *N. eucaryotum* hits to specific variants (e.g. H2A.Z) in the revised manuscript. Note, however, that our objective in this manuscript was to establish whether core histones are present or absent from this species, not whether particular variants are present. Our analysis was not geared towards comprehensive identification of all the histone variants in the genome. While we can therefore show that some variants exist in *N. eucaryotum* (e.g. H2A.Z, see below), we cannot claim the reverse, i.e. state with certainty that a given variant is absent.

Reviewer: 1

2. *This is an interesting addendum to the universality of chromatin composition in eukaryotes that resolves an outstanding apparent exception. My only disappointment is that the authors fail to do even the most rudimentary bioinformatic assessment of the histones they uncover. For example, nearly all eukaryotes have at least two paralogs of H2A, one that is an ortholog of H2A.Z and one or more that is a paralog of H2A/H2A.X. Peculiarly, *Nanochlorum eucaryotum* and its relative *Chlorella variabilis* appear to have only a single H2A. Are these H2As identifiable as H2A.Z-like or H2A.X-like? H2A.X in other organisms is readily identifiable by its C-terminal phosphorylation motif SQE Φ , where Φ is a hydrophobic residue, often F or Y. H2A.Z can be identified not only by conserved residues, especially the extended acidic patch, but even in the divergent H2As of trypanosomes by a one amino acid insertion in Loop1 and a one amino acid deletion in the C-terminal docking domain relative to other H2As. Is the *Nanochlorum* H2A identifiable as either of these conserved paralogs? It would be of interest to know whether it is missing one or both of them.*

We agree with the reviewer that a closer – if preliminary (see comment to the editor above) - look at histone alignments might both further support our original conclusions and also reveal peculiarities of *N. eucaryotum* (if any). We now provide such alignments as Figure S1, focusing on a limited number of eukaryotes where the link between amino acid differences and function is well understood (Human, *D. melanogaster*, *S. cerevisiae*) and two algal species for phylogenetic context (*C. reinhardtii*, *C. variabilis*). Even though the reconstructed *N. eucaryotum* transcript does not cover the entire H2A sequence and misses most of the C-terminal acidic patch, it is nonetheless clearly identifiable as an H2A.Z variant. As expected, it clusters most closely with a *C. variabilis* sequence (EFN59774.1). This does not mean, however, that *N. eucaryotum* only encodes a single H2A paralog. We do not expect our first-path transcriptome assembly to be exhaustive and include all variants

that are encoded in the genome. In fact, taking another look at *C. variabilis* (this time simply looking at Uniprot rather than baiting with the *N. eucaryotum* H2A fragment) we find a number of annotated H2A protein, including two that carry the SQE(F/Y) motif highlighted by the reviewer as a hallmark of H2A.X (Figure S1). We think it likely that a *N. eucaryotum* genome sequence or a further improved transcriptome assembly would similarly reveal the presence of more than one H2A variant. Our analysis was not geared towards being comprehensive and we did not dig deeper into the transcriptome assembly because our objective in this study was not to assess the presence/absence of specific paralogs but rather to demonstrate the presence of histones in general. We discuss these caveats, which naturally also apply to all of the other core histones, in the revised manuscript.

3. *Similarly, Nanochlorum and Chlorella have two H3s each. Unicellular eukaryotes often have an H3.3-like paralog, and most have a cenH3 paralog. While functional assays for cenH3 are presumably beyond the scope of the present manuscript, bioinformatic criteria (long tail, longer Loop 1, and only about 50% identity to conventional H3s) can identify putative cenH3s. This is relevant to “slaying unicorns” because Zahn reported no sign of mitosis or meiosis, whereas a putative cenH3 would argue for the likelihood of these processes.*

As we do for H2A, we now also provide an alignment for H3 that allows a preliminary assignment of putative H3 orthologs to specific variant classes. Based on this alignment, we can say the following: one of the *N. eucaryotum* hits (TRINITY_DN82006_c0_g2_i1) is identical to one of the *C. variabilis* orthologs (EFN50997.1). Both have the highest similarity to metazoan H3.1, and carry the SVM (here SVL) motif rather than the AIG motif characteristic of H3.3. The other H3 candidate (TRINITY_DN79798_c0_g1_i4) is considerably more divergent – although still very similar to *C. variabilis* (EFN57349.1) – and clusters with cenH3 homologs in other species on a neighbour-joining tree (not shown), suggesting that *N. eucaryotum* does in all likelihood encode a cenH3 ortholog.

4. *As a side note, Zahn also reported a lack of microtubules identifiable by inhibitors, and transcripts for tubulins are likely present in the transcriptomes the authors have assembled.*

Using the same domain scanning approach employed for the histones, we searched the assembled transcriptome for homologs to tubulin (Pfam domain PF00091.25). Here too, we retrieved hits with clear homology to other eukaryotic tubulins. As this is not the focus of the paper, we did not investigate this further. However, we now do mention this result in passing.

5. *Having three H2Bs for a single H2A is unusual, and while it may be impossible to assess from bioinformatics what their roles are, It would be interesting to see in an alignment how they differ from each other. Similarly the divergent H4s or H4-like histones are unusual in that H4 is ordinarily one of the most strongly conserved proteins known. Alignments might reveal interesting features, or add to the mystery of their specialization. All four classes of histones typically have conserved sites of modification, especially lysines that can be acetylated, methylated or ubiquitylated.*

Are these conserved residues present in the various paralogs the authors have discovered?

As mentioned in the original manuscript, we only recovered fragments of transcripts with homology to H4. Whereas all other H4 homologs we consider here are very conserved, including the H4 ortholog in *C. variabilis*, the *N. eucaryotum* fragments exhibit a relatively large number of amino acid differences. Given the unusual divergence, it is possible that these fragments might not in fact represent canonical H4 (even though it affiliates with canonical H4 in HistoneDB) but instead belong to a different protein or expressed pseudogene, while transcripts of the canonical H4 have, for unknown reasons, escaped detection. Importantly, however, the MNase digest unequivocally suggests that nucleosomes are being formed and that therefore a functional H4 ortholog must be present.

- 6. Are there one or more H1 paralogs? These questions can presumably be answered relatively easily with the data in hand, and would make for a more interesting paper. Comparison with Chlorella and the other green algal transcriptomes that the authors have assembled might offer some insight into the evolution of Nanchlorum's unique histone complement.*

We find evidence in the transcriptome for two distinct H1 paralogs. As we do not discuss linker histones in the main manuscript we have not considered H1, which are often much more divergent, further.

- 7. Figure 1 Legend: It would be helpful if the authors could elaborate a bit on what they mean by "classical multidimensional scaling". What are the dimensions, or how should I understand this?*

We have revised the legend to briefly clarify what classical multidimensional scaling is and does.